# Precision and temporal dynamics in heading perception assessed by continuous psychophysics

**Björn Jörges**[ID]*, **Ambika Bansal, Laurence R. Harris**[ID]

Center for Vision Research, York University, North York, Canada

* bjoern_joerges@hotmail.de

**Data Availability Statement:** Data are available on Open Science Foundation: https://osf.io/vb9xs

**Funding:** The author(s) received no specific funding for this work.

## Abstract

It is a well-established finding that more informative optic flow (e.g., faster, denser, or presented over a larger portion of the visual field) yields decreased variability in heading judgements. Current models of heading perception further predict faster processing under such circumstances, which has, however, not been supported empirically so far. In this study, we validate a novel continuous psychophysics paradigm by replicating the effect of the speed and density of optic flow on variability in performance, and we investigate how these manipulations affect the temporal dynamics. To this end, we tested 30 participants in a continuous psychophysics paradigm administered in Virtual Reality. We immersed them in a simple virtual environment where they experienced four 90-second blocks of optic flow where their linear heading direction (no simulated rotation) at any given moment was determined by a random walk. We asked them to continuously indicate with a joystick the direction in which they perceived themselves to be moving. In each of the four blocks they experienced a different combination of simulated self-motion speeds (SLOW and FAST) and density of optic flow (SPARSE and DENSE). Using a Cross-Correlogram Analysis, we determined that participants reacted faster and displayed lower variability in their performance in the FAST and DENSE conditions than in the SLOW and SPARSE conditions, respectively. Using a Kalman Filter-based analysis approach, we found a similar pattern, where the fitted perceptual noise parameters were higher for SLOW and SPARSE. While replicating previous results on variability, we show that more informative optic flow can speed up heading judgements, while at the same time validating a continuous psychophysics as an efficient method for studying heading perception.

## Introduction

Optic flow is a major cue to the perception of heading [1, 2], among others including vestibular [3], proprioceptive [4] and tactile [5] cues. It is therefore not surprising that more informative optic flow (e.g., faster simulated self-motion, a larger visual field, or a higher density of optic flow-inducing features in the environment) is generally linked to an improved (i.e., more

**Competing interests:** The authors have declared that no competing interests exist.

precise) perception of heading [6–10]. Here we are defining heading as the direction of movement relative to the head–that is without a rotational component.

Several different classes of models including Bayesian Observer models [11], computer-vision-based models [12] and biologically inspired models [13] have been used to model heading perception [14]. Our present study is motivated by an untested prediction made specifically by the Drift Diffusion Model (DMM) [15–21]. The DMM posits that observers accumulate evidence from a stimulus over time until enough evidence has been acquired to make a perceptual decision. In the case of heading perception, such an accumulation of evidence would imply that more informative optic flow should speed up perceptual heading decisions–a prediction that neither Genova & Stefanova [16] nor Zeng [21] test in their respective studies.

Traditionally, research into heading perception has–like most research investigating perception–employed trial-based methods where participants react to concrete units of stimulus. More recently, researchers have begun using participants' continuous responses to continuously changing stimuli, a paradigm called Continuous Psychophysics [22–27]. Examples of tasks studied using Continuous Psychophysics include the tracking of a randomly moving dot on a 2D surface [22, 23], the perception of simulated self-motion speed from optic flow [24, 25], or eye-movements [26], but any continuous stimulation coupled with a continuous response is, in principle, suitable to be studied in a continuous psychophysics paradigm. This set of methods has the advantage of a) being more naturalistic and ecologically valid because real-life, unlike many traditional experiments, is distinctly *not* chopped up into discrete chunks of perception and action, b) yielding more data per participant time because, effectively, every recorded frame of participant responses corresponds to a data point, which in turn allows the detection of small effects that cannot be detected in trial-based approaches, and c) it allows better study of the temporal dynamics of perceptual phenomena in a more naturalistic setting.

One major advantage of traditional trial-based experiments is that data analysis is often straight-forward and well-understood. Given that the field of continuous psychophysics is still emerging, this is not (yet) the case here. While some simple methods have been proposed in the seminal Bonnen et al. paper [22], there tends to be a focus on less intuitive, complex Kalman filter-based analysis methods [24, 25]. Kalman filters are extensions of Bayesian Observer Models which posit an internal state variable that keeps track of a perceptual variable (such as the current heading direction). This state variable is updated in a Bayesian fashion according to the sensory evidence and the history of changes in the stimulus (as prior). More precise sensory estimates bias the updating of the internal state variable towards the immediate sensory input and a more precise prior biases the update towards the sensory history. It is important to note that, while Kalman filters is the only way to estimate sensory uncertainty, they do so at the cost of buying into the assumptions underlying this category of model, which may hinder adoption of continuous psychophysics. There is, therefore, value in exploring some of the alternative ways to analyze this type of data that allow for a lower specificity in the conclusions drawn, but also require fewer theoretical commitments. Specifically, we will expand on the cross-correlogram analysis employed by Bonnen et al. [22], which allows to assess statistically how fast and how precisely participants react to changes in the stimulus, and we propose a Linear Mixed Model-based analysis that can be useful when a study aims to address variability in performance only.

To our knowledge, two studies [28, 29] have assessed aspects of heading perception using continuous (tracking) paradigms. Both exposed participants to optic flow with a (expanding and contracting) focus of expansion with a focus of expansion that was changing over time and found that a large majority tracked the focus spontaneously with their gaze, i.e., without even being explicitly instructed to do so. [28] further reported that tracking tended to be more

accurate when optic flow was more informative (higher coherence, higher contrast, or higher speed); influences of these manipulations on how fast participants responded to changes in the focus of expansion were not reported.

This paper has two goals: first, it serves as a proof-of-concept for the study of heading perception using Continuous Psychophysics. To this end, we aim to reproduce established findings that more informative optic flow leads to more precise judgements [6–10] by manipulating simulated self-motion speed and the density of the features in an environment providing optic flow. We further use the inherently temporal nature of continuous psychophysics to assess whether more informative optic flow speeds up sensory processes relating to the perception of heading. Finally, this paper presents two ways of analyzing data collected from a task requiring a continuous updating of heading judgements: (1) a simple Cross-Correlogram (CCG) analysis which we combine with bootstrapping to assess population-wide statistical differences between conditions as well as (2) a Kalman filter-based approach tailored to our type of data.

## Methods

### Pre-registration

The pre-registration for this study can be found on OSF (https://osf.io/4s9xg). Please note that this pre-registration also mentions an earlier version of this experiment where we used a starfield instead of the visual environment described below. During data collection, it became clear that the stimulus was too ambiguous for participants to reliably perform the task, leading to the exclusion of about half of the participants tested and extremely high noise in the performance of the remaining participants. We therefore decided against reporting the results of this earlier experiment and only report the experiment with the improved, less ambiguous environment (labelled as "Experiment 2" in the pre-registration).

### Participants

We tested n = 30 participants (12 men, mean of 20 years, standard deviation of 1.1 years; 16 women, mean of 21.6 years, standard deviation of 7.5 years; one declined to declare their gender, 25 years old, and one declared their gender as "not listed", 19 years old), recruited from the York University psychology undergraduate participant pool. Due to the type of study (a methodological proof-of-concept), we did not conduct a power analysis to determine sample size. All participants completed a written consent form and received course credit for their participation. The study was conducted in accordance with the principles of the Declaration of Helsinki and was approved by the York University Office for Research Ethics. Data collection started on March 1$^{st}$, 2024 and ended on April 12$^{th}$, 2024.

### Apparatus

We used an Alienware laptop with 16 GB RAM, an Intel Core i7-9750H CPU (2.60 GHz), and an NVIDIA GeForce RTX 2060 and an VIVE Pro EYE head-mounted device (with a field of view of 110˚ and a resolution of 1440 × 1600 per eye and a 90 Hz refresh rate) to present the stimulus. The stimulus was programmed in Unity (2021.3.13f). The Unity project (https://osf.io/yskgd) and the executable (https://osf.io/6fr5t) can be downloaded from Open Science Foundation. Participants responded using a Logitech G Extreme 3D PRO Joystick, which the Unity program recorded at a rate of 90 Hz. The data were analyzed using R version 4.3.2 [30].

## Stimulus

Participants were immersed in a virtual reality environment that included a white ground plane with planters to provide optic flow (see Fig 1). On each block, they experienced 90 s of visual self-motion on one of nine randomly pre-generated paths through this environment. Observers were always facing in the same direction, i.e., no observer rotation was simulated. The paths were pre-generated using R according to the following three rules: (1) the participants always started moving forwards (coded as 180˚). (2) Every 200 ms, a change in direction was chosen randomly from a distribution with a mean of 0 and a standard deviation of 20˚. (3) If the next 200 ms of motion would have brought the participants outside of a pre-defined circular boundary, 180˚ was added to the new chosen angle of self-motion such that they went in exactly the opposite direction. This was done such that future variations of the same experiment could be conducted on a MOOG motion platform with a limited range of motion.

The R script used to generate these paths can be accessed here (https://osf.io/8af2t, on OSF) and Fig 2A illustrates one of the 9 pre-generated paths. The remaining panels of Fig 2 show the simulated heading angle (B), the change in heading angle (C) and the distribution of changes in heading angle across one 90 s run (D). While experiencing this visual self-motion, participants were asked to continuously indicate the direction in which they thought they were moving using a joystick. In order to minimize head movements, we used a chin rest that was adjusted to each participant's sitting height. No specific instructions were given regarding eye-movements.

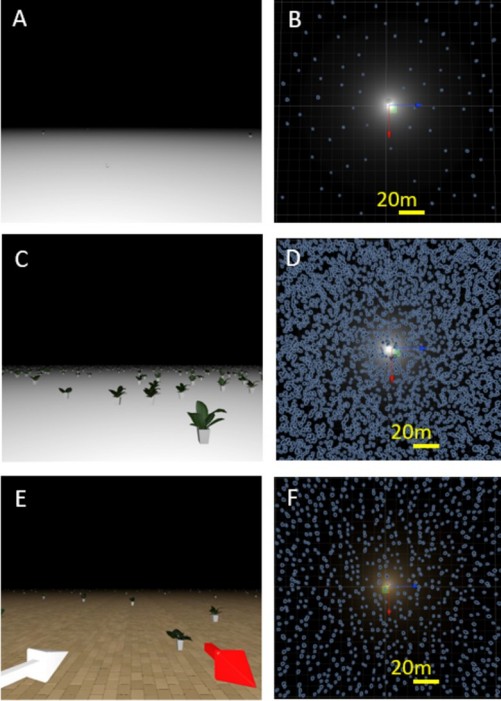

**Fig 1.** A, C, E. Screenshots in the SPARSE condition, DENSE condition, and Training, respectively. Please note the few plants fairly far away from the observer in the DENSE condition (A). B, D, F: Birds-eye view of the scene illustrating the distribution of plants over the area in the SPARSE condition, the DENSE condition, and the Training, respectively. The yellow line provides a scale (one line = 20 simulated meters). Videos of both trainings and all conditions can be found on OSF (SPARSE-SLOW: https://osf.io/ak6pe; SPARSE-FAST: https://osf.io/uc68p; DENSE-SLOW: https://osf.io/23ejp; DENSE-FAST: https://osf.io/yfumk; Training 1: https://osf.io/rk9t4; Training 2: https://osf.io/kfc9s).

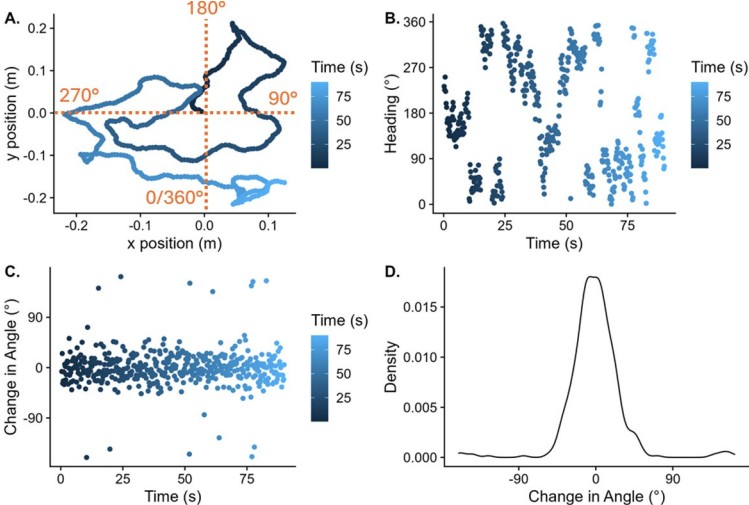

**Fig 2.** A. Simulated x and y position of the participant over the course of a 90 s run (colour-coded, in which the colour gets lighter as time progresses). B. Simulated heading of the participant over the course of one run. Please note that 360˚ = 0˚ for the purposes of this experiment. C. The change in heading angle from one step to the next over the course of one run. Changes in angle of just below +/- 180˚ can occur when the path would have gone past the virtual borders of the experimental area, in the case of which 180˚ were added to/subtracted from the next step to steer the path away from the border. D. The distribution of changes in heading angles across one whole 90 s run, reflecting the normal distribution from which each next step in the random walk was chosen. The fat tails correspond to the angles just below/above +/- 180˚ visible in C.

We manipulated the optic flow our participants received by varying the speed of self-motion (SLOW vs. FAST) and the density of the plants serving to provide optic flow (SPARSE–see Fig 1A; vs. DENSE–see Fig 1C). For the SPARSE condition, we randomly simulated 100 plants in an area of 10,000 m^2 see Fig 1B for a birds-eye view, with a minimum distance of 15 m between each plant in order to avoid clumping. In the DENSE condition we simulated 5,000 plants in the same area (see Fig 1D for a birds-eye view), with a minimum distance of 1 m between them. The plants were generated randomly in the beginning of each run such that the plant placement was different for each condition. The self-motion speed for the SLOW condition was simulated at 0.4–0.6 m/s (changing randomly every 200ms) and for the FAST condition at 2–3 m/s. Eye-height (simulated at 1.3 m to match an average person's sitting eye-height), plant size (simulated at a height of 0.6 m), and self-motion speeds were chosen such that they could be experienced in real-life situations. For purposes of replication, please consult the Unity project (uploaded on OSF: https://osf.io/yskgd) for all details.

## Procedure

Before the four main sequences, participants practiced the task twice on a separate training path. First, they completed a 90 s run in which two arrows (see Fig 1E) indicated the actual direction in which they were moving (a red arrow towards the bottom right of their visual field) and the direction in which they were pointing the joystick (a white arrow to the left). In a second run, only the red arrow indicating their response direction was present. For the training, we chose an intermediate value of 1,000 plants with a minimum distance of 5 m and self-motion speeds between 1 m/s and 1.5 m/s were used (Fig 1F for a birds-eye view). For the training, we also added a tiled ground texture in order to make the perception of self-motion (rather than world motion) even more obvious. This texture was removed for the main experimental condition because it would have provided too obvious cues to heading direction and

potentially induce ceiling effects that would obscure the effects of our manipulations. After completing the training, they were presented with four sequences randomly chosen out of the nine pre-generated trajectories without any arrow being present. Each of these trajectories was presented in one of the four experimental conditions (SPARSE-SLOW, SPARSE-FAST, DENSE-SLOW, DENSE-FAST) in a random order. Overall, the experiment took no more than 30 minutes to finish.

## Data analysis

Fig 3 shows a typical example of the presented heading and the response heading for one participant and condition. For data analysis, we first computed the smallest angle between the shown heading direction and the response heading direction on each frame. We then used a variation of the Cross-Correlogram (CCG) analysis to determine the time lag between the shown heading direction and participant response for which the mean difference between them was the lowest: that is, we offset the presented heading angle from the response angle on any given frame by 0 to 3 s in steps of 5 frames at 60 Hz, i.e., approximately 50 ms and computed the smaller of the two possible angles between them. Then, we took the mean absolute error between the offset presented heading angle and the response angle across a whole run to obtain the error for each time lag (separately for each participant and condition) In order to compare participants and conditions, we then standardized the data by dividing each value by the maximum difference for that participant and condition, which yielded values that could range between 0 and 1. Note that these are ratios of absolute errors, i.e., this number could never be below 0. Fig 4 shows an example of this graph. The point of interest is the deepest "valley" for each condition; the vertical line on each panel indicates the time lag where the difference between shown heading and participant responses was the lowest for any given condition. The horizontal line indicates how small this difference was for this time lag relative to the largest difference for any given condition and participant. Please note that, typically, Cross-Correlogram analyses for continuous psychophysics designs use the change in the input/output variables over time (e.g., the speed of the finger in comparison to the speed of the simulated blob in [22]) rather than the input/output variables themselves. This approach can eliminate unwanted variability due to differences in the stimulus across conditions. In our case, however, since the presented heading only changes every 200 ms, it would force us to average performance within these 200 ms windows in order to effectively compare change in the stimulus to change in the participant response. This would lower our statistical power considerably by condensing 12 frames of data down into one number. Further, by randomly assigning the pre-generated trajectories to each participant and condition, we eliminate the risk that any differences between the conditions could be due to differences in the presented trajectories.

For an outlier analysis, we excluded all participants where the time lag of the least error between stimulus and response was on the borders of the interval for which we calculated the correlations between stimulus and response ([0 s; 3 s]), which indicated that they had not performed the task properly. Values outside of this criterion can generally only occur if participants answered randomly or aligned the joystick with the direction of optic flow rather than the direction of self-motion. This criterion excluded 2 participants (out of a total of 30) from our dataset.

**Bootstrapped cross-correlogram analysis.** In order to assess population-wide statistical significance, we used a bootstrap approach. We performed 1000 repetitions of the Cross-Correlogram (CCG) analysis outlined above over sub-samples of our data each of which contained 25% of the frames recorded for half of the participants. Each sub-sample of participants and

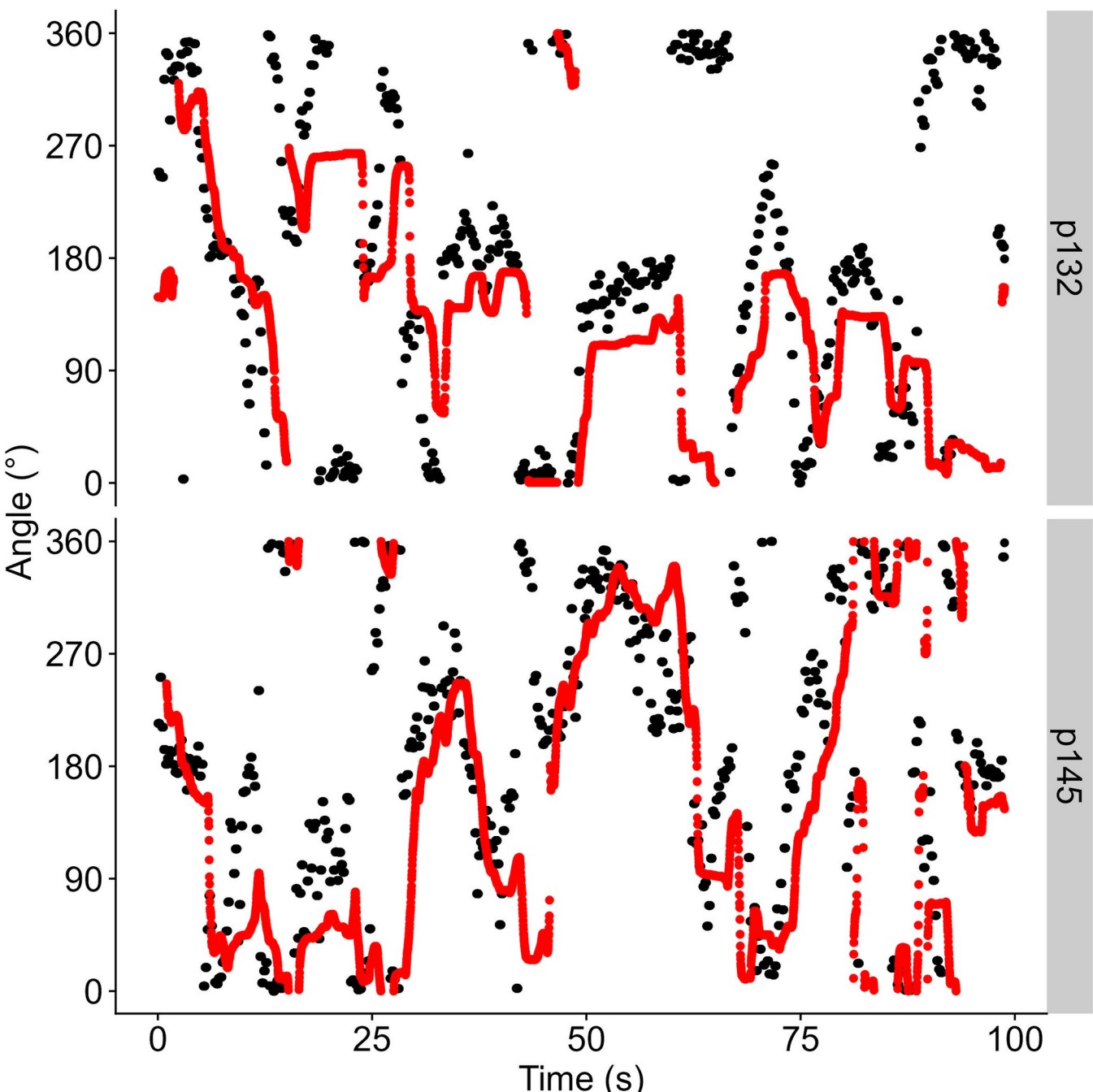

**Fig 3. Stimulus (black) and participant response (red) for two participants (p101, an average participant, and p126, a high performer) for the condition DENSE–FAST.** One dot represents one frame of recorded data.

data points per participant and condition was chosen randomly on any given iteration. Using the Cross-Correlogram analysis, we obtained the minimum difference between stimulus and response as well as the time lag for which this minimum difference was obtained, separately for each participant and optic flow condition. We then used a linear mixed model (LMM; using the lmer function from the lme4 package [31] for R [30]) of the following structure to

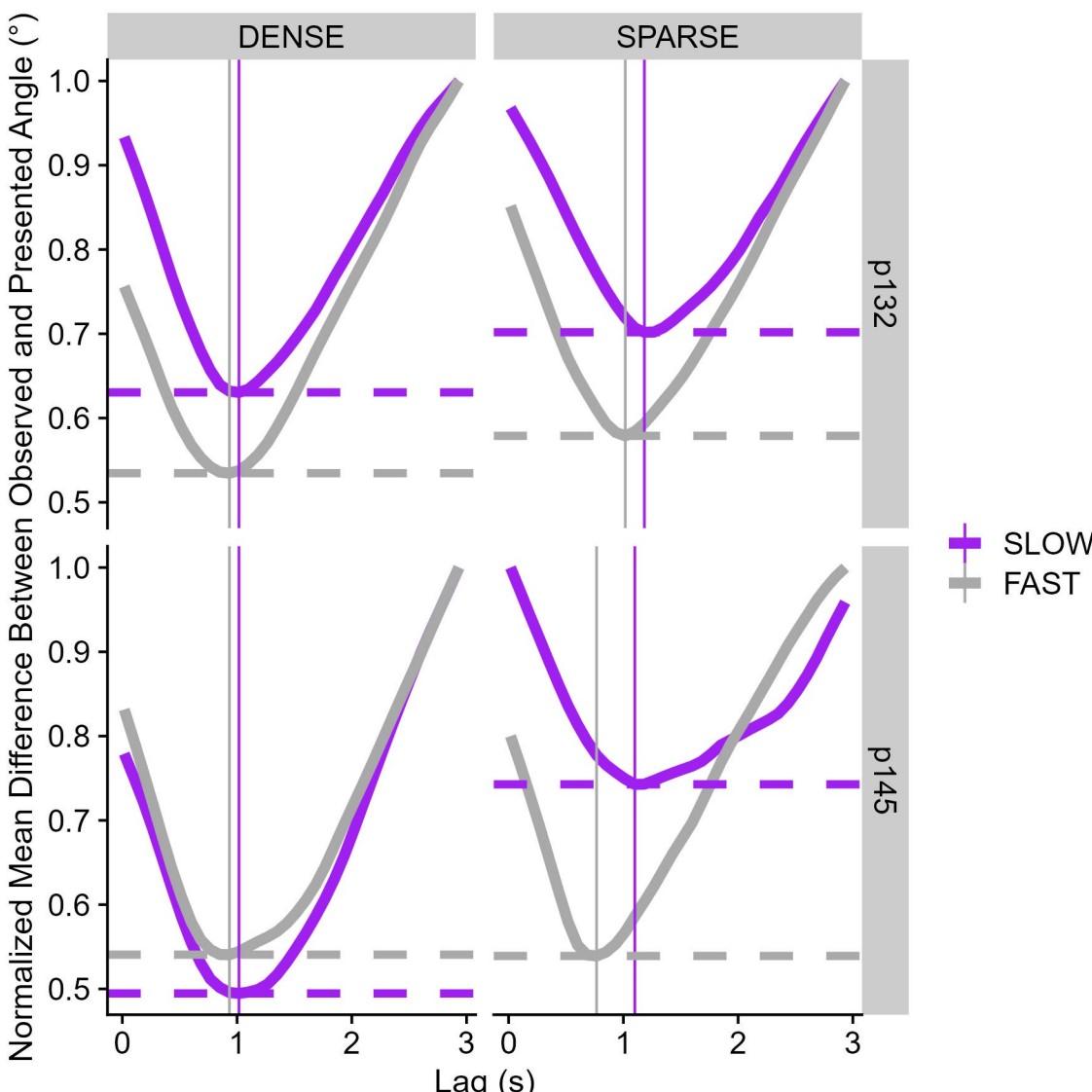

**Fig 4. Normalized mean difference between stimulus and responses (y axis) for a range of lags (x axis) for one average participant (p101) and one high performer (p126), separately for density condition (horizontal panels) and speed condition (color-coded where turquoise = fast and red = slow).**

estimate the difference between SPARSE and DENSE and SLOW and FAST respectively:

$$\text{Minimum Difference} \sim \text{Speed} + \text{Density} + (\text{Speed} + \text{Density} \mid \text{Participant}) \qquad (1)$$

And we fitted the following model for the best time lag:

$$\text{Time Lag of Least Error} \sim \text{Speed} + \text{Density} + (\text{Speed} + \text{Density} \mid \text{Participant}) \qquad (2)$$

Using this approach, we obtained 1000 estimates for the difference contrasts between SLOW and FAST and SPARSE and DENSE for both dependent variables. The 2.5th and 97.5th percentile of these difference contrasts constitute the 95% confidence interval.

We also fitted the above LMMs to the whole data set to obtain the overall (population-wide) regression coefficients.

**Direct estimation of performance variability parameters.** In cases in which the only dependent variable of interest here is how precisely the response matched the stimulus (i.e., when the time lag of least error between stimulus and response is irrelevant), we can forgo the manual bootstrap described above and use linear mixed modelling combined with the bootstrap from the confint function from base R to assess statistical differences. Here, we used the absolute difference between stimulus and response at the time lag of least error (for each participant and condition) on each recorded frame as dependent variable, and the same fixed effect and random effect structure as above:

$$\text{Absolute Error at Time Lag of Least Error} \sim \text{Speed} + \text{Density} + (\text{Speed} + \text{Density}|\text{Participant}) \tag{3}$$

The regression coefficients for Speed and Density then indicate how precisely participants were able to match the stimulus with their response in comparison for each level Speed and Density level.

**Kalman filter.** Following Bonnen et al. (2015), we also modeled our data using a Kalman Filter in order to fit a parameter capturing the participants' sensitivity to the stimulus in the different conditions. In a simplified manner, Kalman filters model optimal observers that use a representation of the change in the stimulus (i.e., the mean and the standard deviation of the distribution, $Q^{0.5}$, from which each new heading direction is chosen) as a prior. The model combines this prior with immediate sensory information (where the mean is taken to be the observed heading direction and the standard deviation, $R^{0.5}$, is fitted to the data, capturing sensory noise in the organism) as likelihood, weighted by their relative reliability, to simulate updating the perceived heading in any given moment. The Kalman gain K, which captures this weighting, can be computed based on the posterior variance P, which, for a Bayesian weighting of prior and likelihood distributions, depends on the variance of the likelihood, R, and the variance of the prior, Q, in the following manner:

$$P = \frac{Q}{2} * \left( \left( 1 + 4 * \frac{R}{Q} \right)^{0.5} - 1 \right) \tag{4}$$

To obtain the Kalman gain, the posterior variance P is then combined with the prior variance Q and the variance of the likelihood R:

$$K = \frac{Q + P}{Q + P + R} \tag{5}$$

The participant response at t is then modelled as perceived heading at t-1 (as indicated by their response, $\hat{x}_{t-1}$) updated by a fraction (determined by the Kalman gain K) of the presented heading angle ($y_t$):

$$\hat{x}_t = \hat{x}_{t-1} + K * (y_t - x_{t-1}) \tag{6}$$

Using the results from the CCG analysis, which showed that it took 1s or more for participants to update their heading response following changes in the stimulus, we used a lag of 1s between stimulus and response.

We fitted the standard deviation of the likelihood, $R^{0.5}$, using maximum likelihood estimation. Based on the fact that the residuals across one run (perceived heading angle scaled by K minus presented heading angle scaled by k; $k * \hat{x}_{t-1} - k * y_t$)) are expected to be normally distributed with a mean of 0 and a standard deviation of $K^2/R$ [22], we minimized the negative log likelihood that these residuals were drawn from a normal distribution with these parameters using the mle function from base R. Please note that Bonnen et al. report maximizing the

likelihood of the Kalman Filter itself; however, their implementation (https://github.com/kbonnen/BonnenEtAl2015_KalmanFilterCode/blob/master/negLogLikelihoodr.m), on which our analysis is based, uses the fitting procedure outlined in the beginning of this paragraph. The fitted standard deviation of the likelihood indicates the participants' sensory sensitivity to the stimulus and can be compared across conditions.

You can find an implementation of all analyses on OSF (https://osf.io/vb9xs).

### Predictions

**CCG parameters.** We expect the time lag of least error (i.e., the time lag at which the absolute error between stimulus and response) to be lower the more informative the presented optic flow is (i.e., lowest for the DENSE-FAST condition and highest for the SPARSE-SLOW condition). Similarly, we expect the peak correlation to be highest for the DENSE-FAST condition and lowest for the SPARSE-SLOW condition.

**LMM analysis.** We expect the absolute error between stimulus and response at the time lag of least error to be lower for the conditions with more informative optic flow (i.e., for higher speeds and a denser arrangement of planters).

**Kalman filter.** We expect the Kalman Filter sensory noise parameters to be lower for conditions with more informative optic flow.

### Results

Fig 3 illustrates an example of raw data for one condition and two participants and Fig 4 shows the CCG plots for the same two participants, while Fig 5 summarizes the extracted Time Lags of Least Difference (A.) and the Least Error between Observed and Response Angle (B.) across all participants.

### Bootstrap

We found evidence in favor of all of our hypotheses: FAST (relative to SLOW) and DENSE (relative to SPARSE) led participants to respond faster to changes in direction (by about 27.1% for FAST relative to SLOW, and by about 14.4% for DENSE relative to SPARSE), as evidenced by significantly lower time lags of least difference. Equally, FAST (relative to SLOW) and DENSE (relative to SPARSE) lowered the normalized least error between stimulus and response heading direction significantly (by 15.5% for FAST relative to SLOW, and by about 9.3%), i.e., responses were less variable. We calculated the percentages based on the intercepts and difference contrasts (as reported in Table 1).

### Regular linear mixed model

Using the regular linear mixed model analysis, we found that FAST led to less variable responses than SLOW (by 11.3°, 95% CI = [8.7°;14°]) and that DENSE led to less variable responses than SPARSE (by 6.9°, 95% CI = [4.8°; 9.3°]). The intercept for this model was 54.3°. That is, FAST lowered the absolute error between stimulus and response by 20.2% relative to SLOW, and by about 12.7% for DENSE relative to SPARSE.

### Modelling

We used the Kalman filter fitting procedure outlined in the Data Analysis section to fit the sensory noise parameter. The distributions of these fitted parameters for Experiment 1 can be found in Fig 6: Results closely matched those for the effects of speed and density on the time

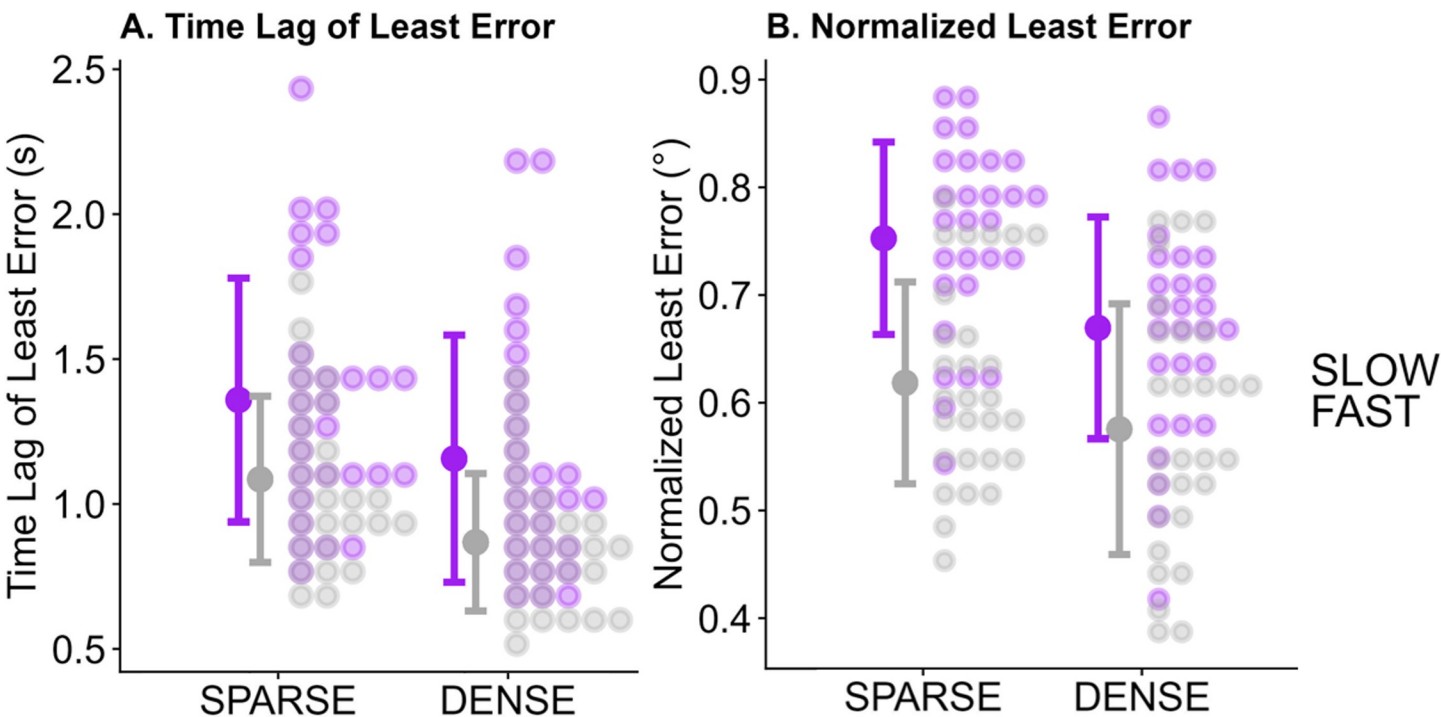

**Fig 5.** Distributions of the Time Lag of Least Error (A) and Normalized Least Error between Observed and Response Angle (B) for density (x axis) and speed (color-coded) conditions. Each dot represents one participant. The large dots represent the mean across all participants and the error bars correspond to +/- 1 standard deviation.

lag of least error and variability in performance; SLOW led to a higher perceptual noise parameter than FAST and SPARSE led to a higher perceptual noise parameter than DENSE.

## Discussion

Overall, we found strong evidence for our hypothesis that more informative optic flow leads to more precise and faster heading responses. All of our predictions were confirmed in the data: more informative optic flow (both through faster simulated self-motion and denser optic flow) led to both less variable and more precise responses as well as to faster reactions in response to changes in direction, as evidenced by all three analyses (bootstrap, LMM, Kalman filter).

**Table 1. Difference contrasts and confidence intervals for the bootstrap analysis.**

|  | *Difference Contrast* | *Confidence Interval (Lower Bound)* | *Confidence Interval (Upper Bound)* | *Significance* |
|---|---|---|---|---|
|  | *Time Lag of Least Difference* | | | |
|  | *(Intercept = 1.18s)* | | | |
| FAST (vs. SLOW) | **-0.32s** | **-0.44s** | **-0.19s** | * |
| DENSE (vs. SPARSE) | **-0.17s** | **-0.34s** | **-0.08s** | * |
|  | *Normalized Least Error (Variability in Performance)* | | | |
|  | *(Intercept = 0.76)* | | | |
| FAST (vs. SLOW) | **-0.11** | **-0.14** | **-0.08** | * |
| DENSE (vs. SPARSE) | **-0.07** | **-0.09** | **-0.03** | * |

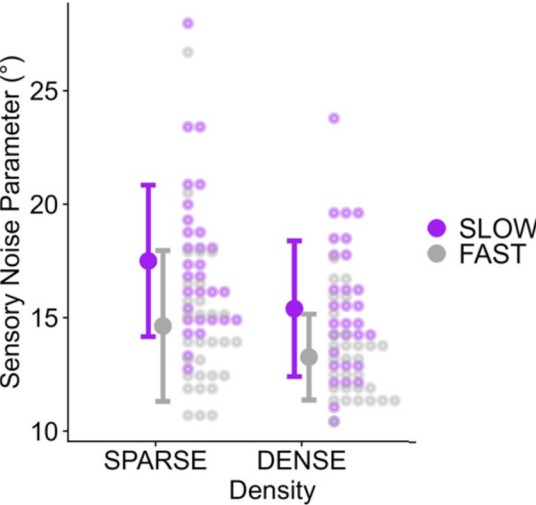

**Fig 6. Distributions of the Kalman filter sensory noise estimates (i.e., the standard deviation of the normal distribution that represents the noise in perceiving the heading direction in any given moment) for density (x axis) and speed (color-coded) conditions.** Each dot represents one participant. The large dots represent the mean across all participants and the error bars correspond to +/- 1 standard deviation.

## More informative optic flow leads to more precise heading responses

Given previous studies in this area [6–10], there was little surprise in finding that more informative optic flow (as manipulation via self-motion speed and density of features in the environment in our experiment) led to more precise heading judgements.

## More informative optic flow leads to faster reactions in responses to changes in heading direction

While current models of heading perception (e.g., the drift diffusion models described in the introduction; [16, 21]) clearly predict that more informative optic flow should lead to faster heading judgements, this had, to our knowledge, not been demonstrated empirically. Our study provides clear evidence that this is, indeed, the case. One interesting direction that our study opens up for future modelling study is the adaptation of drift diffusion models for heading perception for continuous paradigms such as ours.

## Continuous psychophysics is an efficient way to study heading perception

Finally, this study confirms that novel continuous psychophysics methods are well suited to study heading perception, as evidenced by our replication of earlier findings on the relationship between optic flow and variability in heading judgements. This adds to several successful uses of continuous psychophysics to study a range of perceptuo-motor phenomena [22–27].

While continuous psychophysics reproduces this central finding in heading perception, it can do so with a considerably lower time investment by the participants: 4 runs of 90 seconds per participants were enough to produce reliable per-participant estimates for the time lag of least difference as well as their normalized least difference between stimulus and responses. While we did test a comparably large number of participants (n = 30), additional (exploratory) analyses we performed (see Appendix A, B and C in S1 Appendix), showed that these results are fairly robust against dropping either participants (Appendix A in S1 Appendix) or including only parts of the trajectories (Appendix C in S1 Appendix). A common theme was that

analyzing the effect of a manipulation on variability in performance (via the LMM analysis used in the main paper) boasted very high amounts of statistical power and robustness, while the bootstrap analysis (necessary if the effect of a manipulation on the Lag of Least Difference, i.e., the responsiveness of participants to changes in direction, needs to be analyzed) was more susceptible to dropping participants (Appendix A in S1 Appendix). The Lag of Least Difference was also, overall, most vulnerable to randomly dropping parts of the trajectory (Appendix C in S1 Appendix). Finally, we found that performance, both in terms of the Lag of Least Difference (responsiveness) and in terms of the Normalized Least Difference (variability in performance) evolved across the course of a run. While correlations in performance between quarters of a run improved over the course of the run, not even the correlation between the third and the fourth quarters exceeded 0.5, indicating fairly high within-participant variability (Appendix B in S1 Appendix).

Overall, continuous psychophysics is clearly an exciting and useful tool to study perceived heading. Based on our exploratory analysis, there are several recommendations we can make for researchers who want to employ this method:

- Continuous Psychophysics is an excellent tool to study both overall variability in participant responses (e.g., using the LMM analysis suggested in this paper) or sensory noise more specifically (using a Kalman filter-based approach). Our results here were extremely robust, even when conducting the analysis over sub-samples of the data (see Appendix A in S1 Appendix).

- Continuous Psychophysics enables us to study the temporal dynamics of participant responses more naturalistically, but doing so successfully requires larger sample sizes than the study of variability alone.

- Researchers should keep in mind that performance can evolve and develop over the course of a run, and it kept doing so throughout the whole duration of the runs we used in this experiment (90s). It can therefore be useful to increase the length of a run or have the same participants perform several runs in the same condition. This is easily possible as data collection is extremely fast and, anecdotally, our participants were much more engaged in this task than in traditional psychophysical tasks.

While the three analyses we have used in this paper are generally in agreement, it is important to point out similarities and differences between them. The Kalman filter analysis commits to a specific Bayesian framework in which the participant updates a heading estimate on each time step according to the relative reliabilities of 1) prior information on the change in the stimulus and 2) immediate sensory information, i.e., the likelihood. This commitment allows us then to obtain a direct estimate of the sensory uncertainty the participant experiences. The other two analyses (the bootstrapped cross-correlogram analysis and the Linear Mixed Model analysis) require fewer commitments but also allow to draw much less specific, and perhaps more superficial, conclusions: Strictly speaking, these analyses only determine that exposing our participants to more informative optic flow decreases how long it takes them to adjust their motor response and as well as the variability in their response; i.e., no mechanistic conclusions can be drawn about the stage at which this manipulation affects performance. The two main differences between the bootstrapped cross-correlogram analysis and the Linear Mixed Model analysis are that the Linear Mixed Model analysis only allows to assess variability in performance, but not the time course of this perceptuo-motor process, and that its statistical power appears to be somewhat greater (see Appendix A in S1 Appendix). A further caveat that applies to all three of our analyses is that the motor component is likely responsible for a higher share of the overall variability in continuous psychophysics tasks than in trial-based tasks, and

it has been pointed out that modelling this motor component is particularly important in continuous psychophysics [24, 25, 27, 32].

Lastly, it is important to point out two limitations. First, we performed our analysis on the dependent variable itself (i.e., the heading angle) rather than its first temporal derivative (i.e., the change in heading over time), which is the standard for CCG analyses. We made this choice because our stimulus only changed every 200 ms rather than on each frame. To use the temporal derivate, we therefore would have to average participant performance over these 200 ms time windows, which would severely decrease statistical power. It is our understanding that other authors (e.g., [22] have opted for the temporal derivative to minimize the effects of the differences between stimulus trajectories on experimental outcomes. We have minimized this impact by randomizing the trajectories between the experimental conditions, making it unlikely that our results were caused by differences in the stimulus. A second limitation is that the sensory noise estimates obtained in our Kalman filter analysis are generally greater than discrimination thresholds (typically 1–10˚, e.g. [33] would lead us to expect. This may be due to the fact that we don't separate out motor noise in our analysis, or an artifact of our fairly simple environment. A third possibility is we used an unusually large range of heading deviations angles (equally distributed in all directions) whereas lower thresholds are generally reported for heading directions close to the straight-ahead (down to ±1–2˚ around straight ahead; Gu et al., 2010). These effects should, however, affect all conditions equally and therefore not interfere with the interpretation of our results.

## Conclusions

In this paper, we successfully validated the use of Continuous Psychophysics for the study of heading perception. We further replicated earlier findings that more informative optic flow leads to more precise heading judgements and showed that more informative optic flow also speeds up heading judgements (as implied by, e.g., Drift Diffusion Models).

## Open science

All data and code can be found on OSF (https://osf.io/vb9xs), as well as the Unity project (https://osf.io/yskgd) used to generate the executables (https://osf.io/6fr5t).

When analyzing the data, we performed several bug fixes in the pre-registered analysis script, none of which introduced substantive changes to the analysis. In response to reviewer feedback, we performed some bug fixes in the Kalman filter analysis and now fit it using a Maximum Likelihood Approach (rather than the pre-registered Root Mean Square Error-based fitting method). You can view the original script here (https://osf.io/pafq8) for comparison.

The pre-registration can be accessed here (https://osf.io/4s9xg).

## Supporting information

**S1 Appendix.**
(DOCX)

## Author Contributions

**Conceptualization:** Björn Jörges, Ambika Bansal, Laurence R. Harris.

**Data curation:** Björn Jörges, Ambika Bansal.

**Formal analysis:** Björn Jörges.

**Funding acquisition:** Laurence R. Harris.

**Investigation:** Björn Jörges, Ambika Bansal.

**Methodology:** Björn Jörges, Laurence R. Harris.

**Project administration:** Björn Jörges.

**Resources:** Björn Jörges.

**Software:** Björn Jörges.

**Supervision:** Laurence R. Harris.

**Validation:** Björn Jörges.

**Visualization:** Björn Jörges.

**Writing – original draft:** Björn Jörges.

**Writing – review & editing:** Björn Jörges, Ambika Bansal, Laurence R. Harris.

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
