## [Decision Letter · Decision Letter 0]

2 Jun 2024

PONE-D-24-15554Precision and Temporal Dynamics in Heading Perception assessed by Continuous PsychophysicsPLOS ONE

Dear Dr. Jörges,

Thank you for submitting your manuscript to PLOS ONE. After careful consideration, we feel that it has merit but does not fully meet PLOS ONE’s publication criteria as it currently stands. Therefore, we invite you to submit a revised version of the manuscript that addresses the points raised during the review process.

Two experts reviewed your manuscript. Although they both find the topic timely and the paper well-written, they have significant concerns about the methods and analyses. These primarily refer to the implementation of the Kalman model and associated analyses, which you should carefully address. Please note that this invitation for resumbission does not guarantee acceptance for publication in PLoS ONE, as both reviewers will be invited to re-review the manuscript. 

We look forward to receiving your revised manuscript.

Kind regards,

Dimitris Voudouris

Academic Editor

PLOS ONE

Journal Requirements:

2. Your abstract cannot contain citations. Please only include citations in the body text of the manuscript, and ensure that they remain in ascending numerical order on first mention.

Reviewers' comments:

Reviewer's Responses to Questions

**Comments to the Author**

1. Is the manuscript technically sound, and do the data support the conclusions?

Reviewer #1: No

Reviewer #2: Yes

2. Has the statistical analysis been performed appropriately and rigorously? 

Reviewer #1: Yes

Reviewer #2: Yes

3. Have the authors made all data underlying the findings in their manuscript fully available?

Reviewer #1: Yes

Reviewer #2: Yes

4. Is the manuscript presented in an intelligible fashion and written in standard English?

Reviewer #1: Yes

Reviewer #2: Yes

5. Review Comments to the Author

Reviewer #1: In this paper, the authors aim to characterize behavioral responses to changes in the optic flow under various conditions and measure these responses with a continuous psychophysics methodology. This approach allows the authors to obtain parameters like precision much faster than classical single-response methods.

In general, the paper is well written, and the topic is very timely and interesting. However, I have serious concerns regarding the stimuli's temporal structure and its suitability for the Kalman analysis, whose implementation details are completely absent in the paper.

As an additional general comment, although it is very appreciated that people share code and data, the way it is done in this case is very misleading. There appear to be two experiments, while there is only one described in the manuscript. There is no README file to help find the information. For example, I was unable to find the code for the Kalman analysis.

Abstract:

Current models predict faster processing for faster and denser optic flow (OF). However, the focus has been on the accuracy of heading from OF rather than speed. The drift-diffusion model (DDM) presented in the introduction is not very representative of the research in heading from optic flow.

We validate a novel continuous psychophysics paradigm by replicating the effect of the speed and density of optic flow on precision. However, the precision found by the authors is well above previous results, and a direct comparison with more classical approaches is needed.

Main Manuscript:

L7-8: Sensory input (likelihood): It would be preferable to use the term "sensory evidence" and leave "likelihood" for the parameter being optimized in the Kalman (Kalman likelihood) to avoid confusion.

L10-12: This is odd. The only way to obtain a direct measure of observational uncertainty is to use the Kalman filter or linear quadratic Gaussian (LQG) models as in Straub et al. (2022). Unfamiliarity cannot be a justification for not using a Kalman-based approach.

L13-14: In this context, the authors should include citations from continuous psychophysics work (Knoll et al., and others) that test the tracking of the focus of expansion (FOE), which is a critical cue to recover heading from optical flow.

Stimulus Section: Here I have a major concern. The process noise has a standard deviation (SD) of 20 degrees, which is not justified. Such a large variability is well above the precision required to control heading by recovering the FOE in real-life situations. Second, the change is made every 200 ms. Therefore, the structure is not a random walk, meaning that during 200 ms, observers can integrate flow consistent with a single direction. This is not a problem per se unless the Kalman model is adapted to these process dynamics. Unfortunately, there is no information in the paper concerning the implementation of the Kalman model.

Data Analysis Section: - L19-20: The cross-correlogram (CCG) is run on angular positions (heading and response), which is not the usual way. The authors should have used the speed of changes to compute the CCG. However, the stimulus structure (changes every 200 ms) might interfere with the computation of the CCG, which assumes a new pair of values in each frame.

Data Analysis Section - Kalman Filter: There is a total lack of information about the model, and I could not find the implementation in the OSF. The reported sensory noise is between 50 and 100 degrees, which are very large values. Given that the necessary information to evaluate the Kalman filter is missing, I am very unconfident that the analysis was conducted properly.

Reviewer #2: In the present manuscript, the authors explore the use of continuous psychophysics as a method to investigate the perception of heading directions. In an experiment with a virtual reality setup, subjects indicated their perceived heading direction while their actual heading direction changed on a random walk. The speed of self-motion and the optic flow density were manipulated in a within-subjects design. Using cross-correlograms and a Kalman filter model, the authors conclude that more informative optic flow, both due to density and due to faster self-motion, leads to more accurate and faster perception of heading direction.

The manuscript is well written and easy to follow. The exposition of the relevant concepts (optic flow, processing speed and continuous psychophysics) in the introduction is clear. The study is a nice application of continuous psychophysics to heading perception as a new stimulus variable. The goals of the study are laid out clearly in the final paragraph of the introduction and the results are structured in a way that makes it easy for the reader to relate the findings to these goals. The main result of more precise and quicker heading direction perception with better optic flow seems to be supported by the presented results, although some concerns about the methods remain, which are discussed below.

The manuscript is a prime example of open science. All data have been made available and the code needed to reproduce the experiment and analyses is accessible. I checked the provided R code for the data analysis, which ran without errors and produced figures looking identical to those in the paper. A pre-registration is referenced and the reasons for deviations from the pre-registered procedures are indicated.

Since I am not an expert on heading perception, I cannot write with confidence on the significance of the novel finding of faster processing of heading direction with more informative optic flow, so my following comments will focus on the data analysis procedures.

1. To better understand the reasons for the deviation from the pre-registration, I have a question about the results from Experiment 1, which are described in the pre-registration document for experiment 2 (https://osf.io/um7cg). The document states that the authors "found no significant differences between any of the conditions for any of the experimental conditions for either of the two dependent variables". Table 1 in the document, however, indicates that the confidence interval of the difference contrasts between the fast and slow conditions exclude zero in for both the time lag and the normalized least error. Please clarify the statistical procedures used to assess significance.

2. The description of the Kalman filter analysis in the manuscript is very sparse. While it provides a good intuition about the model, some equations describing how the parameters were set or fit to the data would be helpful. After reading and executing the code, I have a couple minor concerns. Neither of these should invalidate the qualitative results in the present study, but they should be addressed in the interest of a proper use of the method and better interpretability of the values. a) Judging from the equations, which seem to follow the notation in Bonnen et al. (2015), the parameter Q should be the variance of the target random walk. In the manuscript (section "Stimulus"), the standard deviation of the target random walk is given as 20°, which would result in a variance of Q = 400°^2. Changing Q to this value in the code does not qualitatively change the effects, but it does change the concrete values for the sensory noise parameter. b) Following the previous point, the parameter R in the notation by Bonnen et al. (2015) should be the variance of the sensory noise in degrees^2, but the y axis of Figure 7 states the values as degrees. If we want to interpret the values as sensory uncertainty in the units of the experiment, reporting actual standard deviations is preferred. c) The procedure in the function KalmanFilter is based on random model simulations and is used to minimize the absolute error between the KF estimates and the subject's responses. The use of random simulations introduces variability in the obtained estimates due to the fitting and could be addressed by instead using the likelihood function in Appendix B of Bonnen et al. (2015) or in their code on https://github.com/kbonnen/BonnenEtAl2015_KalmanFilterCode/blob/master/negLogLikelihoodr.m.

3. The CCG analysis seems non-standard. Typically, the cross-correlation between the velocity of the target and the velocity of the cursor is used, as in Mulligan (2002); Mulligan et al. (2013); Bonnen et al. (2015) and many other applications of tracking tasks. In the present manuscript, the authors compute the CCG on position instead of velocity, which might confound the dynamics of the target with the dynamics of the response. Furthermore, they use the mean absolute error between the target and response instead of the correlation. Is there a reason for these deviations from previous procedures? Since the target movement was the same across all conditions, I suspect that this will not systematically bias any of the conditions more than the others. However, to enable a comparison with other continuous psychophysics studies, I would like to additionally see the velocity CCGs and their statistics.

As mentioned earlier, none of these methodological concerns are likely to affect the qualitative effects on which the conclusions are based. This is an interesting application of continuous psychophysics to a naturalistic task, which will be a valuable addition to the literature once the minor points above are addressed.

As a final point, I want to comment on the comparison between the Kalman filter and the CCG as different analysis approaches for continuous psychophysics. The introduction states that "reliance on Kalman filter modelling may hinder adoption of continuous psychophysics, either because researchers may not be familiar with this approach or because they prefer not to buy into the assumptions underlying a Kalman filter". This is not a good reason against the use of a method, because there is no model without assumptions. On the contrary, I would view it as an advantage of Kalman filters and other probabilistic approaches that they transparently state their assumptions. On the other hand, it is less clear which assumptions underlie the interpretation of the peak or the lag of a CCG as a measure of perceptual precision or processing speed. Furthermore, multiple studies have emphasized the need to model the motor system in addition to a perceptual model like the Kalman filter (Ambrosi et al., 2022; Straub et al., 2022; Falconbridge et al., 2023), since these factors also affect behavior. If the goal is to obtain quantitative estimates from continuous psychophysics that can be clearly attributed to perceptual, cognitive, and motor processes (like sensitivity and criterion in SDT), these factors need to be taken into account. Besides these theoretical considerations, there are also empirical differences between the measuremements obtained from the different analysis methods. In the discussion, the authors say: "Continuous Psychophysics is an excellent tool to study the precision of participant responses, be it using a Kalman filter-based approach or the LMM analysis suggested in this paper." While the qualitative effects in the present study are the same between the KF and the CCG summary statistics, it looks like they do capture slightly different aspects of participants' behavior. If the authors would like to provide recommendations on the preferred analysis methods for future continuous psychophysics studies, a more detailed examination of the differences between the results obtained from the different analysis approaches and their robustness, would be appropriate.

References:

- Bonnen, K., Burge, J., Yates, J., Pillow, J., & Cormack, L. K. (2015). Continuous psychophysics: Target-tracking to measure visual sensitivity**.** *Journal of vision*, *15*(3), 14-14.

- Mulligan, J. B. (2002). Sensory processing delays measured with the eye-movement correlogram. *ANNALS-NEW YORK ACADEMY OF SCIENCES*, *956*, 476-478.

- Mulligan, J. B., Stevenson, S. B., & Cormack, L. K. (2013, March). Reflexive and voluntary control of smooth eye movements. In *Human Vision and Electronic Imaging XVIII* (Vol. 8651, pp. 223-244). SPIE.

- Ambrosi, P., Burr, D. C., & Cicchini, G. M. (2022). Ideal observer analysis for continuous tracking experiments. *Journal of Vision*, *22*(2), 3-3.

- Straub, D., & Rothkopf, C. A. (2022). Putting perception into action with inverse optimal control for continuous psychophysics. *Elife*, *11*, e76635.

- Falconbridge, M., Stamps, R. L., Edwards, M., & Badcock, D. R. (2023). Continuous psychophysics for two-variable experiments; A new “Bayesian participant” approach. *i-Perception*, *14*(6), 20416695231214440.

6. PLOS authors have the option to publish the peer review history of their article (what does this mean?). If published, this will include your full peer review and any attached files.

Reviewer #1: No

Reviewer #2: No

---

## [Author Response · Author response to Decision Letter 0]

8 Jul 2024

Our responses to the reviewers contain figures and tables, so please refer to the work document we have uploaded.

---

## [Decision Letter · Decision Letter 1]

6 Aug 2024

PONE-D-24-15554R1Precision and Temporal Dynamics in Heading Perception assessed by Continuous PsychophysicsPLOS ONE

Dear Dr. Jörges,

Thank you for submitting your manuscript to PLOS ONE. After careful consideration, we feel that it has merit but does not fully meet PLOS ONE’s publication criteria as it currently stands. Therefore, we invite you to submit a revised version of the manuscript that addresses the points raised during the review process. Both reviewers still have concerns about the implementation of the Kalman filter, which should be carefully addressed.

We look forward to receiving your revised manuscript.

Kind regards,

Dimitris Voudouris

Academic Editor

PLOS ONE

Reviewers' comments:

Reviewer's Responses to Questions

**Comments to the Author**

1. If the authors have adequately addressed your comments raised in a previous round of review and you feel that this manuscript is now acceptable for publication, you may indicate that here to bypass the “Comments to the Author” section, enter your conflict of interest statement in the “Confidential to Editor” section, and submit your "Accept" recommendation.

Reviewer #1: (No Response)

Reviewer #2: (No Response)

2. Is the manuscript technically sound, and do the data support the conclusions?

Reviewer #1: No

Reviewer #2: Partly

3. Has the statistical analysis been performed appropriately and rigorously? 

Reviewer #1: No

Reviewer #2: Yes

4. Have the authors made all data underlying the findings in their manuscript fully available?

Reviewer #1: Yes

Reviewer #2: Yes

5. Is the manuscript presented in an intelligible fashion and written in standard English?

Reviewer #1: Yes

Reviewer #2: Yes

6. Review Comments to the Author

Reviewer #1: I appreciate the efforts made by the authors to address my previous concerns. However, there are two issues (one more critical) that still need to be addressed:

a) Model Fit Methodology: Upon reviewing the analysis code, it appears that the fit of the Kalman filter is not based on minimizing the negative log-likelihood as the authors claim in the manuscript, as would be expected in order to obtain an optimal estimate. Instead, the authors have opted to minimize the mean error between the response and the Kalman posterior estimate. This method does not use the square of the error or even the root mean square error (RMSE), which raises concerns about the validity of the estimates. This approach does not typically guarantee that the estimated parameters are the most likely given the data, which could potentially lead to unreliable conclusions. This relates to point b.

b) Estimate Sizes: The parameter estimates provided are unusually large (about 15degs), which seems implausible in the context of the studied phenomena. This issue might suggest a scaling error in the data or model mis-specification. It is crucial that the authors reassess these estimates to ensure they are realistic and reflect the true dynamics of the data. One possibility is that motor noise is being affecting the estimates, but I doubt it. The authors should test how the process noise value is affecting their estimates, but through a true minimization of the neg likelog, which is absent in the paper.

I recommend that the authors re-evaluate these aspects of their methodology to ensure the robustness and validity of their findings.

Reviewer #2: The revised version addresses most concerns I had about the original mansucript, but makes a few of its problems more apparent. In particular, I have two minor points about the description of the KF analysis and the stimulus dynamics. The final point concerns the statements about the assumptions of different models and why I think the authors do not do themselves a favor by making these statements or bringing up the DDM. For each point, I make recommendations for how to address it.

1.The Kalman filter implementation has been fixed and the code now seems to implement a maximum likelihood estimate. The authors have added a bit of description of the model in the paper. Here, I am confused by one particular sentence: "The participant response at t+1 is then modelled as presented heading at t multiplied by the Kalman gain K". This is inconsistent with the version of the KF model of continuous psychophysics by Bonnen et al. (2015), where the subject's response is assumed to be the KF estimate $\\hat x$. This could be clarified by writing out the equations for the KF, i.e. how the belief is updated, and specifying exactly which variables are assumed to correspond to behavior and which to the state of the experiment.

2.In response to comments by both reviewer 1 and me about the non-standard CCG analysis, the authors clarified that a standard velocity CCG analysis was not possible because of the target dynamics in their task, which operates at a time step of 200ms, while the behavioral response has a higher frame rate. This is a limitation of the experimental design of the present study and should be discussed as such.

3.Both reviewer 1 and I criticized the statement in the original version of the manuscript that "reliance on Kalman filter modelling may hinder adoption of continuous psychophysics, either because researchers may not be familiar with this approach or because they prefer not to buy into the assumptions underlying a Kalman filter." The authors have now slightly rearranged that paragraph. They have removed unfamiliarity of researchers with the approach as a reason for not adopting the modeling approach, which is commendable. Still, the revised version states "the cost of buying into the assumptions underlying this category of model" as a reason for not using models based on Bayesian inference and optimal control and that "other analysis methods require fewer theoretical commitments".

I want to push back on this because, in the revised version, both the introduction and discussion feature the drift diffusion model (DDM) as a potential model for modeling perception in continuous tasks. The revised introduction even states that "this study focusses on one particular class of models that has been applied to heading perception: Drift Diffusion Models". This is confusing, because there is no analysis based on DDMs anywhere to be found in the paper. More importantly, though, the DDM also makes assumptions, like any model does. There is typically an algorithmic process of noisy evidence accumulation towards a bound with particular assumptions about the functional form of noises, drift rates, and bounds. If the authors want to criticize one class of models for making assumptions while favoring another class of models, they could start a discussion about why one set of assumptions is preferable over another based on theoretical or empirical considerations. But simply stating that adoption of one class of models is hindered because people "prefer not to buy into the assumptions" is not going to suffice, especially when the present manuscript provides no argument (theoretical or empirical) for why DDMs might be more appropriate for continuous psychophysics tasks. Both the comment about the assumptions of KF and LQG as models of CPP and the statements about the DDM contribute nothing of value to the paper and should be removed, in my opinion. If the authors insist on making comments about the assumptions of different models, I would expect a better justification. Similarly, if the DDM really was a central focus of the paper, I would expect to see it in effect for the task at hand.

7. PLOS authors have the option to publish the peer review history of their article (what does this mean?). If published, this will include your full peer review and any attached files.

Reviewer #1: No

Reviewer #2: No

---

## [Author Response · Author response to Decision Letter 1]

14 Sep 2024

Response to the Reviewers

Reviewer #1: I appreciate the efforts made by the authors to address my previous concerns. However, there are two issues (one more critical) that still need to be addressed:

a) Model Fit Methodology: Upon reviewing the analysis code, it appears that the fit of the Kalman filter is not based on minimizing the negative log-likelihood as the authors claim in the manuscript, as would be expected in order to obtain an optimal estimate. Instead, the authors have opted to minimize the mean error between the response and the Kalman posterior estimate. This method does not use the square of the error or even the root mean square error (RMSE), which raises concerns about the validity of the estimates. This approach does not typically guarantee that the estimated parameters are the most likely given the data, which could potentially lead to unreliable conclusions. This relates to point b.

RESPONSE #1: We followed the procedure outlined by Bonnen et al. (2015), which implements a negative log likelihood estimation. We are assuming that the modelling fitting step the reviewer refers to is the one where we first determine the errors between predicted and observed values (line 64 in Analysis Kalman filter.R). We then determine the negative log likelihood that these values come from a normal distribution with a mean of 0 and a standard deviation of K^2 * R, as Bonnen et al describe it in lines 19 – 22 of their matlab script (https://github.com/kbonnen/BonnenEtAl2015_KalmanFilterCode/blob/master/negLogLikelihoodr.m).

b) Estimate Sizes: The parameter estimates provided are unusually large (about 15degs), which seems implausible in the context of the studied phenomena. This issue might suggest a scaling error in the data or model mis-specification. It is crucial that the authors reassess these estimates to ensure they are realistic and reflect the true dynamics of the data. One possibility is that motor noise is being affecting the estimates, but I doubt it. The authors should test how the process noise value is affecting their estimates, but through a true minimization of the neg likelog, which is absent in the paper.

I recommend that the authors re-evaluate these aspects of their methodology to ensure the robustness and validity of their findings.

RESPONSE #2: We have double-checked our code and corrected a concrete mistake spotted by reviewer #2. This has, however, not lowered the parameter estimates significantly. We are confident in this version of the code, and given how hard the task was for some participants (anecdotally), we believe that these values are sensible. Furthermore, what matters for our conclusions are the differences between the conditions, which are clear, and in line with the results from the CCGs. (See, e.g., Bonnen 2015, who found similarly striking differences in best time lag and peak correlation between their conditions, which were correlated strongly with the fitted Kalman filter sensory noise estimates.) Further, heading discrimination is much less sensitive outside of a +-45° window around the straight-ahead (Gu et al., 2010); only 25% of our frames fell into this more sensitive window. Consequently, since our analysis implicitly averages across all heading angles, the overall sensitivity estimates should therefore be much higher (indicating lower precision) than the 1-2° reported by Gu et al. (2010) for the straight-ahead. We have, nonetheless, noted the reviewers concern in the discussion as readers should be aware of this mismatch between the literature and our study, along with some tentative explanations.

Reviewer #2: The revised version addresses most concerns I had about the original manuscript but makes a few of its problems more apparent. In particular, I have two minor points about the description of the KF analysis and the stimulus dynamics. The final point concerns the statements about the assumptions of different models and why I think the authors do not do themselves a favor by making these statements or bringing up the DDM. For each point, I make recommendations for how to address it.

1.The Kalman filter implementation has been fixed and the code now seems to implement a maximum likelihood estimate. The authors have added a bit of description of the model in the paper. Here, I am confused by one particular sentence: "The participant response at t+1 is then modelled as presented heading at t multiplied by the Kalman gain K". This is inconsistent with the version of the KF model of continuous psychophysics by Bonnen et al. (2015), where the subject's response is assumed to be the KF estimate $\\hat x$. This could be clarified by writing out the equations for the KF, i.e. how the belief is updated, and specifying exactly which variables are assumed to correspond to behavior and which to the state of the experiment.

RESPONSE #3: We have corrected this mistake in the code and now properly conduct the Kalman filter fitting in line with Bonnen et al. (2015). We have also corrected the phrasing in the manuscript and have added some further detail in the modelling section.

2.In response to comments by both reviewer 1 and me about the non-standard CCG analysis, the authors clarified that a standard velocity CCG analysis was not possible because of the target dynamics in their task, which operates at a time step of 200ms, while the behavioral response has a higher frame rate. This is a limitation of the experimental design of the present study and should be discussed as such.

RESPONSE #4: We have added this is in the discussion section.

3.Both reviewer 1 and I criticized the statement in the original version of the manuscript that "reliance on Kalman filter modelling may hinder adoption of continuous psychophysics, either because researchers may not be familiar with this approach or because they prefer not to buy into the assumptions underlying a Kalman filter." The authors have now slightly rearranged that paragraph. They have removed unfamiliarity of researchers with the approach as a reason for not adopting the modeling approach, which is commendable. Still, the revised version states "the cost of buying into the assumptions underlying this category of model" as a reason for not using models based on Bayesian inference and optimal control and that "other analysis methods require fewer theoretical commitments".

I want to push back on this because, in the revised version, both the introduction and discussion feature the drift diffusion model (DDM) as a potential model for modeling perception in continuous tasks. The revised introduction even states that "this study focusses on one particular class of models that has been applied to heading perception: Drift Diffusion Models". This is confusing, because there is no analysis based on DDMs anywhere to be found in the paper. More importantly, though, the DDM also makes assumptions, like any model does. There is typically an algorithmic process of noisy evidence accumulation towards a bound with particular assumptions about the functional form of noises, drift rates, and bounds. If the authors want to criticize one class of models for making assumptions while favoring another class of models, they could start a discussion about why one set of assumptions is preferable over another based on theoretical or empirical considerations. But simply stating that adoption of one class of models is hindered because people "prefer not to buy into the assumptions" is not going to suffice, especially when the present manuscript provides no argument (theoretical or empirical) for why DDMs might be more appropriate for continuous psychophysics tasks. Both the comment about the assumptions of KF and LQG as models of CPP and the statements about the DDM contribute nothing of value to the paper and should be removed, in my opinion. If the authors insist on making comments about the assumptions of different models, I would expect a better justification. Similarly, if the DDM really was a central focus of the paper, I would expect to see it in effect for the task at hand.

RESPONSE #5: We agree that DDMs occupied an outsized space in the introduction and have cut down our mention of them to the minimum required to convey (one of the) motivations for this study: DDMs predict faster judgements for more informative optic flow, a prediction that has not been tested yet.

Further, we have clearly failed to properly express our opinions regarding the assumptions underlying the Kalman filter in the paper. Our intention was never to claim that DDMs make fewer (or even no) assumptions in comparison to a Kalman filter. Rather, we propose the (bootstrapped) CCG-based analysis and the LMM analysis as alternatives that requires fewer assumptions (namely they can be used without assuming a Bayesian updating process) but also allow for less specific conclusions (we can only report the maximum correlations and best time lags obtained via CCGs, or changes in precision via the LMM, but we can NOT draw any conclusions about the uncertainty in the participant’s representation of the stimulus). We have amended these sections to make our comparison clearer.

References

Bonnen, K., Burge, J., Yates, J., Pillow, J., & Cormack, L. K. (2015). Continuous psychophysics: Target-tracking to measure visual sensitivity. Journal of Vision, 15(3). https://doi.org/10.1167/15.3.14

Gu, Y., Fetsch, C. R., Adeyemo, B., DeAngelis, G. C., & Angelaki, D. E. (2010). Decoding of MSTd Population Activity Accounts for Variations in the Precision of Heading Perception. Neuron, 66(4), 596–609. https://doi.org/10.1016/j.neuron.2010.04.026

---

## [Decision Letter · Decision Letter 2]

22 Sep 2024

PONE-D-24-15554R2Precision and Temporal Dynamics in Heading Perception assessed by Continuous PsychophysicsPLOS ONE

Dear Dr. Jörges,

Thank you for submitting your manuscript to PLOS ONE. After careful consideration, we feel that it has merit but does not fully meet PLOS ONE’s publication criteria as it currently stands. Therefore, we invite you to submit a revised version of the manuscript that addresses the minor but important points that are raised by Reviewer 1.

We look forward to receiving your revised manuscript.

Kind regards,

Dimitris Voudouris

Academic Editor

PLOS ONE

Journal Requirements:

Additional Editor Comments (if provided):

Reviewers' comments:

Reviewer's Responses to Questions

**Comments to the Author**

1. If the authors have adequately addressed your comments raised in a previous round of review and you feel that this manuscript is now acceptable for publication, you may indicate that here to bypass the “Comments to the Author” section, enter your conflict of interest statement in the “Confidential to Editor” section, and submit your "Accept" recommendation.

Reviewer #1: All comments have been addressed

Reviewer #2: All comments have been addressed

2. Is the manuscript technically sound, and do the data support the conclusions?

Reviewer #1: Partly

Reviewer #2: Yes

3. Has the statistical analysis been performed appropriately and rigorously? 

Reviewer #1: No

Reviewer #2: Yes

4. Have the authors made all data underlying the findings in their manuscript fully available?

Reviewer #1: Yes

Reviewer #2: Yes

5. Is the manuscript presented in an intelligible fashion and written in standard English?

Reviewer #1: Yes

Reviewer #2: Yes

6. Review Comments to the Author

Reviewer #1: The manuscript has improved clarity in some aspects, but a couple of key issues remain.

First, the type of negative log-likelihood (negloglik) that the authors are minimizing is not based on maximizing the Kalman log-likelihood, but rather on the negloglik of the residuals (i.e., the difference between the response and the posterior). While this is now clear in the text and not technically incorrect, the authors continue to cite Bonnen et al. (2015) as if they are using the same procedure, which is misleading. In Bonnen’s paper, they maximize the log-likelihood of the Kalman itself (equations B15/B16) by fitting the observational noise parameter R , which can be interpreted as visual precision. This is fundamentally different from what the current manuscript is doing. While I still believe this could affect the estimates, I will not pursue this point further. However, the authors should correct this misrepresentation and clearly distinguish between the two approaches, ensuring that Bonnen et al. is cited appropriately.

Second, the manuscript misleadingly presents LMM as a novel method for measuring precision, which is not accurate. The essence of continuous psychophysics (CPP) is to derive a measure of precision, often equivalent to Just Noticeable Differences (JND) or differential thresholds. The manuscript uses the term “precision” loosely in several sections. In the context of sensory processing, precision refers to the inverse of variance or uncertainty, as typically modeled by Kalman Filters and Bayesian approaches (or JND in classical psychophysics). The authors should avoid conflating this concept with what LMM captures. While LMM is appropriate for analyzing condition-based differences in performance, it does not estimate sensory noise or uncertainty as Kalman Filters do. LMMs, as used in the manuscript, are not designed to estimate precision in the context of sensory processing. Instead, they only reveal significant differences between conditions based on absolute error—accuracy would likely be a more appropriate metric here. The claim that LMM offers a complementary method to the Kalman Filter for measuring precision is incorrect and needs to be revised. Clarifying these methodological distinctions would enhance the manuscript’s clarity and main contribution.

I suggest that the authors focus on LMM as a tool for detecting performance differences between conditions based on error or accuracy. The manuscript’s primary focus should shift toward this type of analysis, reframing LMM’s role in understanding performance variation, without suggesting that it measures precision. While this approach may be less exciting than measuring variability or sensory precision, it is a more accurate representation of what is the main contribution of the study.

Finally, the discussion should also be revised to reflect these clarifications.

Reviewer #2: (No Response)

7. PLOS authors have the option to publish the peer review history of their article (what does this mean?). If published, this will include your full peer review and any attached files.

Reviewer #1: No

Reviewer #2: No

---

## [Author Response · Author response to Decision Letter 2]

24 Sep 2024

Reviewer #1: The manuscript has improved clarity in some aspects, but a couple of key issues remain.

First, the type of negative log-likelihood (negloglik) that the authors are minimizing is not based on maximizing the Kalman log-likelihood, but rather on the negloglik of the residuals (i.e., the difference between the response and the posterior). While this is now clear in the text and not technically incorrect, the authors continue to cite Bonnen et al. (2015) as if they are using the same procedure, which is misleading. In Bonnen’s paper, they maximize the log-likelihood of the Kalman itself (equations B15/B16) by fitting the observational noise parameter R , which can be interpreted as visual precision. This is fundamentally different from what the current manuscript is doing. While I still believe this could affect the estimates, I will not pursue this point further. However, the authors should correct this misrepresentation and clearly distinguish between the two approaches, ensuring that Bonnen et al. is cited appropriately.

1. Bonnen et al. (2015) use the approach that we have implemented, as described/justified in a comment in their code https://github.com/kbonnen/BonnenEtAl2015_KalmanFilterCode/blob/master/negLogLikelihoodr.m: [Please see screenshot in Response to Reviewer.doc]

We have now made this explicit in our manuscript.

Second, the manuscript misleadingly presents LMM as a novel method for measuring precision, which is not accurate. The essence of continuous psychophysics (CPP) is to derive a measure of precision, often equivalent to Just Noticeable Differences (JND) or differential thresholds. The manuscript uses the term “precision” loosely in several sections. In the context of sensory processing, precision refers to the inverse of variance or uncertainty, as typically modeled by Kalman Filters and Bayesian approaches (or JND in classical psychophysics). The authors should avoid conflating this concept with what LMM captures. While LMM is appropriate for analyzing condition-based differences in performance, it does not estimate sensory noise or uncertainty as Kalman Filters do. LMMs, as used in the manuscript, are not designed to estimate precision in the context of sensory processing. Instead, they only reveal significant differences between conditions based on absolute error—accuracy would likely be a more appropriate metric here. The claim that LMM offers a complementary method to the Kalman Filter for measuring precision is incorrect and needs to be revised. Clarifying these methodological distinctions would enhance the manuscript’s clarity and main contribution.

I suggest that the authors focus on LMM as a tool for detecting performance differences between conditions based on error or accuracy. The manuscript’s primary focus should shift toward this type of analysis, reframing LMM’s role in understanding performance variation, without suggesting that it measures precision. While this approach may be less exciting than measuring variability or sensory precision, it is a more accurate representation of what is the main contribution of the study.

2. We agree that it is extremely important to keep apart sensory uncertainty as measured by, e.g., the Kalman Filter, and performance differences as measured by alternative methods of analyzing such as our LMM approach. We are keenly aware of this difference and have now made this clearer in the manuscript as follows:

We have changed “precision” to “variability in performance” throughout unless we are talking specifically about sensory uncertainty (as determined by the Kalman filter). We don’t think “accuracy” would communicate our intentions appropriately because, in our understanding, the terms refers to (systematic) biases rather than (random) variability or noise in performance. While there are applications for an investigation into biases, this project was set up to assess variability in performance in the different conditions.

Finally, the discussion should also be revised to reflect these clarifications.

3. We have made the appropriate changes in the discussion as well.

---

## [Decision Letter · Decision Letter 3]

29 Sep 2024

Precision and Temporal Dynamics in Heading Perception assessed by Continuous Psychophysics

PONE-D-24-15554R3

Dear Dr. Jörges,

We’re pleased to inform you that your manuscript has been judged scientifically suitable for publication and will be formally accepted for publication once it meets all outstanding technical requirements.

Kind regards,

Dimitris Voudouris

Academic Editor

PLOS ONE

Additional Editor Comments (optional):

Reviewers' comments:

Reviewer's Responses to Questions

**Comments to the Author**

1. If the authors have adequately addressed your comments raised in a previous round of review and you feel that this manuscript is now acceptable for publication, you may indicate that here to bypass the “Comments to the Author” section, enter your conflict of interest statement in the “Confidential to Editor” section, and submit your "Accept" recommendation.

Reviewer #1: All comments have been addressed

2. Is the manuscript technically sound, and do the data support the conclusions?

Reviewer #1: (No Response)

3. Has the statistical analysis been performed appropriately and rigorously? 

Reviewer #1: (No Response)

4. Have the authors made all data underlying the findings in their manuscript fully available?

Reviewer #1: (No Response)

5. Is the manuscript presented in an intelligible fashion and written in standard English?

Reviewer #1: (No Response)

6. Review Comments to the Author

Reviewer #1: (No Response)

7. PLOS authors have the option to publish the peer review history of their article (what does this mean?). If published, this will include your full peer review and any attached files.

Reviewer #1: No

---

## [Editor Report · Acceptance letter]

2 Oct 2024

PONE-D-24-15554R3 

PLOS ONE

Dear Dr. Jörges, 

I'm pleased to inform you that your manuscript has been deemed suitable for publication in PLOS ONE. Congratulations! Your manuscript is now being handed over to our production team.

Kind regards, 

on behalf of

Dr. Dimitris Voudouris 

Academic Editor

PLOS ONE